# Robust and inducible genome editing via an all-in-one prime editor in human pluripotent stem cells

Youjun Wu [1], Aaron Zhong[1], Mega Sidharta[1], Tae Wan Kim[2], Bernny Ramirez[1], Benjamin Persily[1], Lorenz Studer [2] ✉ & Ting Zhou [1] ✉

Prime editing (PE) allows for precise genome editing in human pluripotent stem cells (hPSCs), such as introducing single nucleotide modifications, small insertions or deletions at a specific genomic locus. Here, we systematically compare a panel of prime editing conditions in hPSCs and generate a potent prime editor, "PE-Plus", through co-inhibition of mismatch repair and p53-mediated cellular stress responses. We further establish an inducible prime editing platform in hPSCs by incorporating the PE-Plus into a safe-harbor locus and demonstrated temporal control of precise editing in both hPSCs and differentiated cells. By evaluating disease-associated mutations, we show that this platform allows efficient creation of both monoallelic and biallelic disease-relevant mutations in hPSCs. In addition, this platform enables the efficient introduction of single or multiple edits in one step, demonstrating potential for multiplex editing. Our method presents an efficient and controllable multiplex prime editing tool in hPSCs and their differentiated progeny.

Prime editing (PE) utilizes a Cas9n fused reverse transcriptase along with prime editing guide RNA (pegRNA) to copy the desired DNA mutation from the programmable RNA template within the pegRNA into the specified genomic location[1]. Prime editing can introduce all types of single-nucleotide polymorphism (SNP) changes, small deletions, or insertions at a specific genome site[1]. Compared to CRISPR-based homology-directed repair (HDR), there are no double-stranded DNA breaks (DSBs) or need for donor templates[2]. Compared to base editing, there is no limitation of editing window or specific SNP types[2–4]. Due to its precision and versatility in genome manipulation, prime editing stands out as an ideal gene editing technology for applications such as disease modeling, gene function studies, and mutation corrections[5–10].

Systematic evaluation of gene function and genetic variants is crucial for the study of normal human physiology and the modeling and treatment of disease. Genome-wide gene perturbation screens, such as CRISPR/Cas9, CRISPRi, and CRISPRa approaches, have proven highly effective for achieving high throughput endogenous gene knockout[11–14], knockdown[15–18], or activation[19–21]. After delivering lentivirus-based sgRNA into Cas9 or dCas9-expressing cells for one week, the DNA cutting or binding efficiency can reach over 90%[11,21]. However, unlike gene perturbation strategies, there is still a lack of inducible and scalable approaches for creating precise endogenous genetic variants. Previous studies have utilized base editors for massively parallel assessment of human variants[22]. However, base editing approaches have limitations in terms of precision and the range of sgRNA targets[23]. While prime editing stands out as an attractive option for achieving high-throughput, precise genetic variant editing, it presents challenges due to its relatively lower editing efficiency compared to other genome manipulation tools.

More advanced systems have been developed in the field of prime editing[24–27]. For instance, PE4 and PE5 were created by disrupting the DNA mismatch repair (MMR) pathway using a dominant negative MLH1 protein that inhibits MMR (MLH1dn) in conjunction with PE2 and PE3 (PE4 = PE2 + MLH1dn; PE5 = PE3 + MLH1dn)[24]. The editing efficiency of PE4 and PE5 can be further enhanced when combined with

[1]The SKI Stem Cell Research Facility, The Center for Stem Cell Biology and Developmental Biology Program, Sloan-Kettering Institute for Cancer Research, 1275 York Avenue, New York, NY, USA. [2]The Center for Stem Cell Biology and Developmental Biology Program, Sloan-Kettering Institute for Cancer Research, 1275 York Avenue, New York, NY, USA. ✉e-mail: studerl@mskcc.org; zhout@mskcc.org

PEmax, a more efficient prime editor with an optimized architecture (PE4max and PE5max)[24]. Our previous study demonstrated that transiently inhibiting the p53 by co-delivering a dominant negative domain of p53 (P53DD) also significantly improves the efficiency of prime editing in hPSCs through inhibition of p53-mediated cellular stress responses)[27].

In this study, we aim to establish a robust and inducible prime editing platform in hPSCs. We initially conducted a comprehensive comparison of the editing efficiency among a series of prime editors and identified the most effective configuration for application in hPSCs. Subsequently, we establish an all-in-one prime editor termed "PE-Plus" by incorporating three key components- PEmax, MLH1dn, and P53DD- that collectively contribute to the highest PE efficiency. To enhance the effectiveness, reliability, and versatility of prime editing in hPSCs, we further develop a universal platform termed "iPE-Plus" by integrating PE-Plus into the *AAVS1* safe harbor locus, using TALEN-mediated gene targeting. This platform enables the inducible expression of the prime editor, making the editing efficiency finely regulated by doxycycline, thereby allowing for adjustable editing efficiency and precise editing in specific cell types. Using this platform, we demonstrate that over 50% of Parkinson's-related N370S mutation in the *GBA* gene and cancer-related L858R mutation in the *EGFR* gene can be achieved after 7 days of doxycycline incubation. This advancement streamlines the process of obtaining single-cell clones carrying the desired mutations without the need to screen many single clones. In addition, this platform increases the proportion of clones with biallelic mutations, which are the type of models often required for the study of linking mutations to gene function. Furthermore, we demonstrate that the iPE-Plus platform allows for the generation of disease models with multiplex mutations through one-step induction in hPSCs. These findings demonstrate the capacity of the iPE-Plus platform for inducible and effective prime editing with remarkable flexibility and reliability.

## Results

### Comparison of current PE tools in human pluripotent stem cells

Multiple prime editing tools have been developed to improve prime editing efficiency in mammalian cells. However, determining the optimal PE tool for use in hPSCs remains uncertain. To address this question, we generated a "H2B-turn-on reporter" for real-time monitoring of prime editing outcomes (Fig. 1a). This reporter construct was generated by introducing an H2B-tdTomato cassette into one allele of *SOX2* locus in the hESC H1 line, where a "C" deletion in the H2B sequence was present. This "C" deletion caused a frameshift between the tdTomato gene and the *SOX2* gene, resulting in the lack of tdTomato fluorescence. Prime editing was applied to reintegrate the "C" nucleotide within H2B, allowing tdTomato to be expressed in-frame with *SOX2*, thereby activating fluorescence (Fig. 1b). Therefore, the proportion of tdTomato-positive cells reflects the efficiency of prime editing. In most cases, the additional nicking sgRNA significantly enhances prime editing for endogenous loci compared to PE2[1,27]. However, this improvement was not observed in the "H2B-turn-on reporter" system, which showed similar editing efficiency with or without the additional nicking sgRNA (Supplementary Fig. 1). Therefore, no nicking sgRNA was included in the comparison of different conditions in this reporter system. Various prime editing conditions were evaluated based on FACS data (Fig. 1d, e). As expected, PE4 demonstrated enhanced editing efficiency compared to the original PE2, resulting in an increase from 12.2% to 18.0% in tdTomato-positive cells. The utilization of either PE2 or PE4 along with P53DD led to further improvements (23.9% with PE2 + P53DD; 30.3% with PE4 + P53DD) in tdTomato-positive cell population. Notably, the combination of PE4 and P53DD exhibited the highest efficacy among the tested conditions, indicating the additive enhancement resulting from concurrent inhibition of the MMR pathway and P53. By substituting the

PE2 enzyme with PEmax, editing efficiency was synergistically amplified in conjunction with the aforementioned four conditions (18.4% with PE2max, 19.8% with PE4max, 29.2% with PE2max + P53DD, 38.0% with PE4max + P53DD). Moreover, we explored two versions of engineered pegRNA (epegRNA), each containing distinct 3'RNA motifs to enhance pegRNA stability[25]. The combination of either epegRNA, PE4max, and P53DD led to further enhancement in PE efficiency, with more than 55% of cells exhibiting tdTomato fluorescence 48 h after transient transfection (Fig. 1c, d). To confirm our observations using the "H2B-turn-on reporter" system, we performed amplicon sequencing (Miseq) to quantify frequencies of prime editing and byproducts. The addition of P53DD (PE4max + P53DD) or substituting the pegRNA with epegRNA (PE4max + tev or PE4max + tmp) increased prime editing efficiencies of PE4max from 8.4% to 18% ~ 19%. Combining PE4max with P53DD and epegRNA further improved the editing efficiency to 24% ~ 27% (Fig. 1f). Meanwhile, the editing specificity, calculated as the ratio of on-target edits to byproducts, was not compromised (Fig. 1g). Normalization of data from the two experiments, as determined by the fold change of tdTomato-positive cells assessed by FACS (Fig. 1h) and editing frequencies assessed by Miseq relative to PE4max (Fig. 1i), showed high consistency, with approximately two-fold increases when PE4max was combined with either P53DD or epegRNA, and around three-fold increases when PE4max was combined with both P53DD and epegRNA.

To further validate the observation from the reporter system that PEmax, MLH1dn, and P53DD yielded the most effective prime editing, we conducted different types of editing at endogenous genomic loci in hPSCs. The combination with MLH1dn and P53DD significantly improved 2nt deletion at the *HEK3* site to 10.4%, compared with either PEmax (4.2%) or PEmax with MLH1dn (5.7%) (Fig. 1j). In the cases of larger size editing, MLH1dn did not show any increase of editing efficiency when conducting a 30nt deletion (Fig. 1k) or a 34nt "Loxp" insertion (Fig. 1l) at the *HEK3* locus, while the addition of P53DD substantially improved PEmax-mediated 30nt deletion from 3.1% to 12.1% and 34nt insertion from 7.9% to 24.3%. Similar results were observed in the case of a 10nt deletion at the *SOX2* locus, showing 0.56% with PEmax, 0.8% with PEmax+MLH1dn, and 2.5% with PEmax+MLH1dn + P53DD (Fig. 1m). Since the precise deletion efficiencies at the *SOX2* locus were much lower than those at the *HEK3* locus, we applied twin-PE, which is expected to show a higher editing ratio for larger size editing using a pair of pegRNAs on a 40nt deletion at the *SOX2* locus[28]. PEmax and PEmax+MLH1d showed 2.8% and 2.4% ratio of precise 40nt deletion, respectively, while PEmax + MLH1d + P53DD dramatically increased the on-target efficiency to 16.7% (Fig. 1n). This improvement is mainly due to the P53DD, as PEmax combined with P53DD alone showed a similar editing efficiency (15.7%). Taken together, these data indicate that the combination of PEmax, MLH1dn, and P53DD was identified as the most effective PE condition in hPSCs, where MLH1dn and P53DD additively improve the efficiency for smaller edits, and the P53DD component can significantly improve the efficiency for larger edits achieved with the PE platform.

### Generation of All-in-one PE-Plus prime editor

We next aimed to construct a primer editor that incorporated all three components found to yield the highest prime editing efficiency in hPSCs, including PEmax, MLH1dn and P53DD. Our first step was the optimization of the P53DD to achieve a more efficient enhancement of PE. The P53DD that was previously assessed in our study was derived by mouse and consisted of the first 13 codons of mouse p53, followed by its C-terminal region containing three domains: nuclear localization signal (NLS) region, tetramerization domain (TET) and C-terminal negative regulatory domain (CTD)[29,30]. We asked whether a human-derived P53DD would exhibit greater efficiency in human stem cells (Supplementary Fig. 2a). Similar to the construction of the mP53DD plasmid, the human P53DD plasmid (hP53DD) expresses the C-terminal

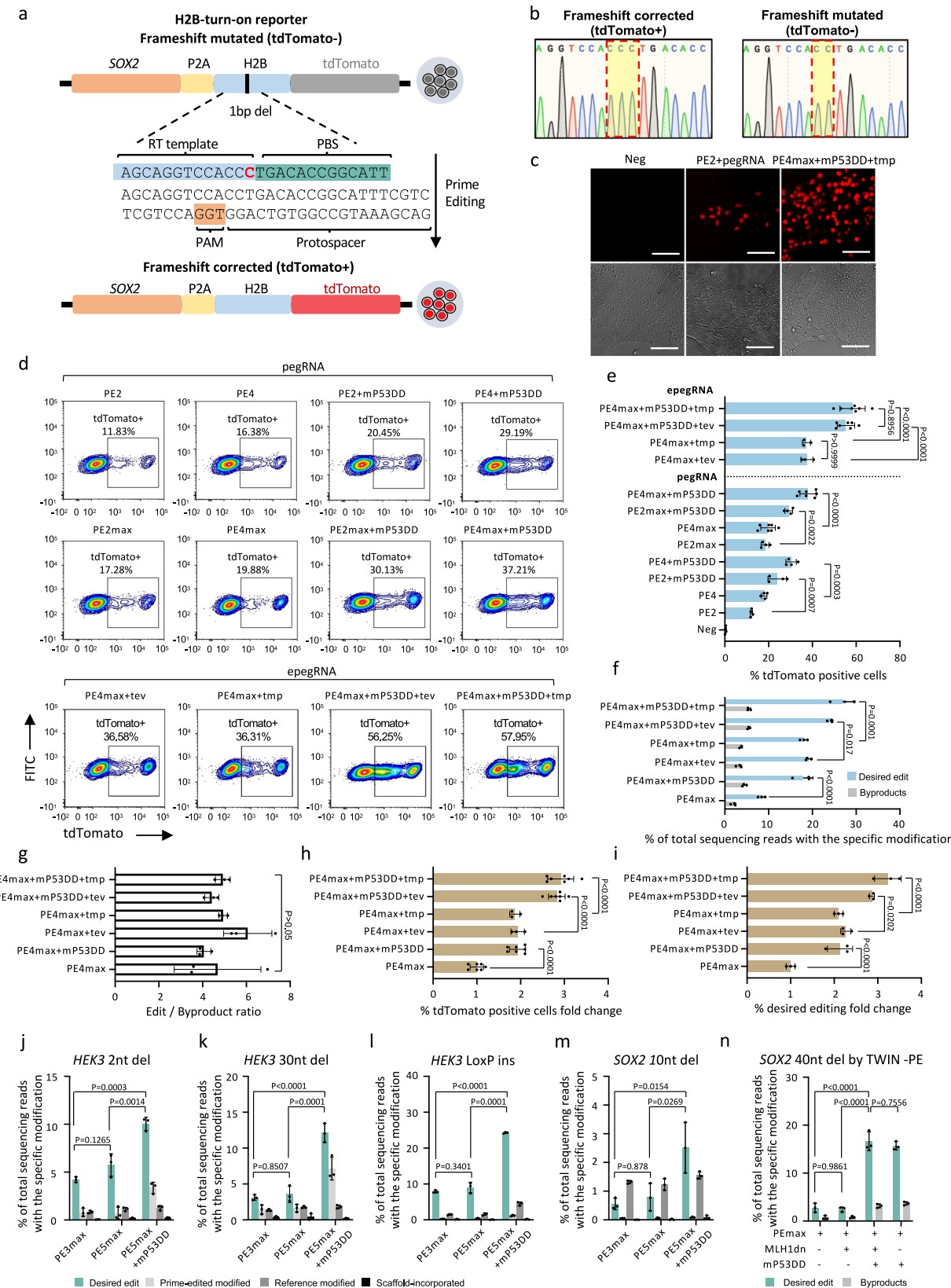

region of human P53, which is essential for the formation of a stable P53/DNA complex[31]. The cloned C-terminal region functions to inhibit P53 by competition for DNA binding. We cloned the full C-terminal region of human P53, consisting of all three domains: the NLS, TET, and CTD. Meanwhile, truncated hP53DD only contained TET and CTD domains, and a variant with only the TET domain, was also cloned for

comparisons (Supplementary Fig. 2b). Using the PE reporter cell line as mentioned above, we assessed PE efficiency in the presence of mP53DD, hP53DD and hP53DD truncations (Supplementary Fig. 2c, d). We observed improvement in PE efficiency across all the P53DD plasmids. Specifically, mP53DD and hP53DD yielded the highest editing efficiency (27.5% with mP53DD and 30.8% with hP53DD), whereas the

**Fig. 1 | Comparison of different prime editing tools in hPSCs using an "H2B-turn-on reporter" cell line. a** Schematic of the H2B-turn-on reporter for evaluating prime editing efficiency. **b** Sequence of the region in the H2B-turn-on reporter cells (right panel) containing a "C" deletion and the sequence after restoration of the "C" by prime editing (left panel). **c** Detection of tdTomato under a fluorescence microscope before (Neg) and 48 h after prime editing with indicated conditions in the reporter cells. Bright-filed images are provided in the lower panel. Scale bar: 10 μm. One of four independent experiments ($n = 4$) is shown. **d, e** Representative FACS plots (**d**) and bar graph (**e**) showing the percentage of tdTomato-positive cells 48 h after electroporation. $n = 4$ independent electroporation reactions for Neg, PE2, PE4, PE2 + mP53DD, PE4 + mP53DD, PE2max, PE2max + mP53DD; $n = 7$ for PE4max, PE4max + mP53DD, PE4max + mP53DD + tev, PE4max + mP53DD + tmp; $n = 3$ for PE4max + tev and PE4max + tmp. **f** Miseq analysis of the desired "C" insertion and the byproduct frequencies 48 h after electroporation with different

prime editing conditions ($n = 3$ independent electroporation reactions). **g** Prime editing outcome purity calculated by edit/byproduct ratio ($n = 3$ independent electroporation reactions). **h** Fold change in the percentage of tdTomato-positive cells under the indicated prime editing conditions relative to PE4max. $n = 7$ independent electroporation reactions for PE4max, PE4max + mP53DD, PE4max + mP53DD + tev, PE4max + mP53DD + tmp; $n = 3$ for PE4max+tev and PE4max + tmp. **i** Fold change in desired "C" insertion frequencies under the indicated editing conditions normalized to that of PE4max ($n = 3$ independent electroporation reactions). **j–n** Miseq analysis of desired and undesired edits of a 2nt deletion (**j**), a 30nt deletion (**k**), a 34nt "Loxp" insertion at the *HEK3* locus (**l**), a 10nt deletion (**m**), and a 40nt deletion (**n**) at the *SOX2* locus with indicated prime editing conditions ($n = 3$ independent electroporation reactions). Data in **e–n** are presented as mean ± S.D. *p*-values were calculated by one-way ANOVA with Tukey's multiple comparison test (**e–n**). Source data are provided as a Source Data file for (**e–n**).

truncated P53DD variants did not perform as effectively as the full-length hP53DD (23.3% with hP53DD-TET-CTD and 18.8% with hP53DD-TET). These results suggested that both mouse and human P53DD improve prime editing efficiency in hPSCs. Furthermore, all three domains within the C-terminal region were found to contribute to the enhancement of PE editing. This can potentially be attributed to the distinct functions of each domain in DNA binding, and the incorporation of all three domains seems to lead to the most effective inhibition of P53. However, a comprehensive understanding of the underlying mechanisms will require further elucidation. The two versions of P53DD were further assessed by targeting more endogenous loci, including the induction of the N370 mutation at the *GBA* locus, the L858R mutation at the *EGFR* locus, the G12C mutation at the *KRAS* locus, as well as a "LoxP" insertion at the *HEK3* locus (Supplementary Fig. 2e). The enhancement effect of the two P53DD versions over PE3 was similar according to the desired edits at all the tested targeting sites (Supplementary Fig. 2f), indicating that either one can be chosen for the PE-Plus system.

Next, we incorporated hP53DD and human hMLH1dn into the PEmax prime editor either through direct fusion or by utilizing linkages in between. Two fusion plasmids were generated on top of PEmax-P2A-hMLH1dn (PE4max), with an additional hP53DD directly fused to N-terminus or C-terminus of PEmax (Fig. 2a). The H2B-turn-on reporter assay showed that the two plasmids with direct fusion of hP53DD reduced the prime editing efficiency, indicating a negative impact of protein fusion on the prime editor (6.7% with N-terminal fusion, 10.8% with C-terminal fusion) (Fig. 2b). In contrast, the plasmid with linkages PEmax-P2A-hP53DD-IRES-hMLH1dn increased editing efficiency compared with PE4max, rising from 17.8% to 28.8%. We further tested the plasmid with two P2A linkages (PEmax-P2A-hP53DD-P2A-hMLH1dn). The data showed no difference in the editing efficiency generated by the plasmids with P2A/IRES or P2A/P2A linkages (Supplementary Fig. 3). We name the robust all-in-one PE editor as PE-Plus (PEmax-P2A-hP53DD-IRES-hMLH1dn).

### Analysis of genome-wide off-target effects induced by PE-Plus

The PE-Plus edited cells and PEmax edited cells were then enriched by isolating the tdTomato positive cells after frame restoration in the "H2B-turn-on reporter". The editing outcomes in the tdTomato-positive cell populations were evaluated by Miseq analysis, which showed that 86% of the sorted cells in either condition exhibited perfect "C" insertion. The rate of unwanted modifications, including "reference modified," "prime-edited modified" "scaffold-incorporated," and "ambiguous," was low (3.05% with PEmax and 2.72% with PE-Plus in total). (Supplementary Fig. 4).

To evaluate the genome-wide safety of the PE-Plus prime editor, we performed whole genome sequencing (WGS) to identify unwanted off-target effects in the genome. Both SNV and indel mutations in the genome were identified by comparing them to the unedited parental cells (Fig. 2d). The number of insertions and deletions was not

increased upon PE-Plus treatment (36 insertions and 57 deletions in PEmax-edited cells; 35 insertions and 38 deletions in PE-Plus-edited cells). Similarly, no increase in SNVs was observed upon PE-Plus treatment (322 in PEmax-edited cells vs. 296 in PE-Plus-edited cells) (Fig. 2e). Among the SNV mutations, the two prime editors resulted in a similar mutation pattern in the genome, as analyzed by the number and relative proportions of SNV mutation types (Fig. 2f). Consequently, these data suggest that PE-Plus improves prime editing efficiency without compromising genome-wide safety.

### Generation iPE-Plus hPSC cells

To achieve more efficient and versatile prime editing in hPSCs, we generated a prime editing platform for the doxycycline-induced expression of PE-Plus. As this inducible-PE platform did not require transfection, editing efficiency would not be affected by plasmid size or cytotoxicity upon plasmid delivery. The iPE-Plus hPSCs were engineered by TALEN-mediated genome editing targeting the safe harbor locus *AAVS1* (Supplementary Fig. 5a, b). One donor plasmid expressing PE-Plus under the control of TRE promoter (TRE-PE-Plus-Hygro) and another donor plasmid expressing M2rtTA under the CAG promoter (CAG-M2rtTA-Neo), were electroporated into hPSCs along with a pair of *AAVS1* TALEN vectors (Fig. 3a). To achieve inducible mutations in the genome, iPE-Plus cells were pre-infected with pegRNA and/or nicking sgRNA lentivirus. The intended edits would then be installed into the genome upon doxycycline treatment (Fig. 3a).

To provide a preliminary assessment of iPE-Plus hPSCs, we utilized the "H2B-turn-on reporter" system described above (Fig. 3a, b). Rapid and efficient inducible expression of the prime editor was observed in three selected clones after doxycycline treatment, as demonstrated by qPCR (Supplementary Fig. 5c). The iPE-Plus reporter cells were then infected with either pegRNA or epegRNA lentivirus, followed by doxycycline treatment for different durations to active tdTomato (Fig. 3b). The tdTomato positive cells was checked at various time points following doxycycline addition. In the presence of pegRNA, the population of tdTomato + cells increased to 35.1% at day 2 and further to 69.0% at day 4. The increase then slowed down after day 4, gradually reaching 85.8% at day 8 (Fig. 3c–e). In cells infected with epegRNA lentivirus, tdTomato was switched on more rapidly, with 77.5% at day 2, reaching a plateau after day 4 (Fig. 3c–e). Furthermore, to access induced prime editing in an endogenous gene, we applied this platform with an epegRNA lentivirus to insert a "TGA" stop codon into the *SOX2* gene in the heterozygous H1-SOX2-tdTomato reporter line, thereby silencing tdTomato expression (TGA-insertion reporter) (Supplementary Fig. 6a.) FACS data and fluorescence images demonstrated the gradual switch-off of tdTomato over 12 days. Approximately 23.7% of cells became tdTomato negative after 2 days of dox treatment, increasing to 65.1% at day 4, and eventually reaching 90% at day 10 (Supplementary Fig. 6b–d). These results collectively demonstrate inducible and time-dependent prime editing by the iPE-Plus platform, controlled by doxycycline treatment.

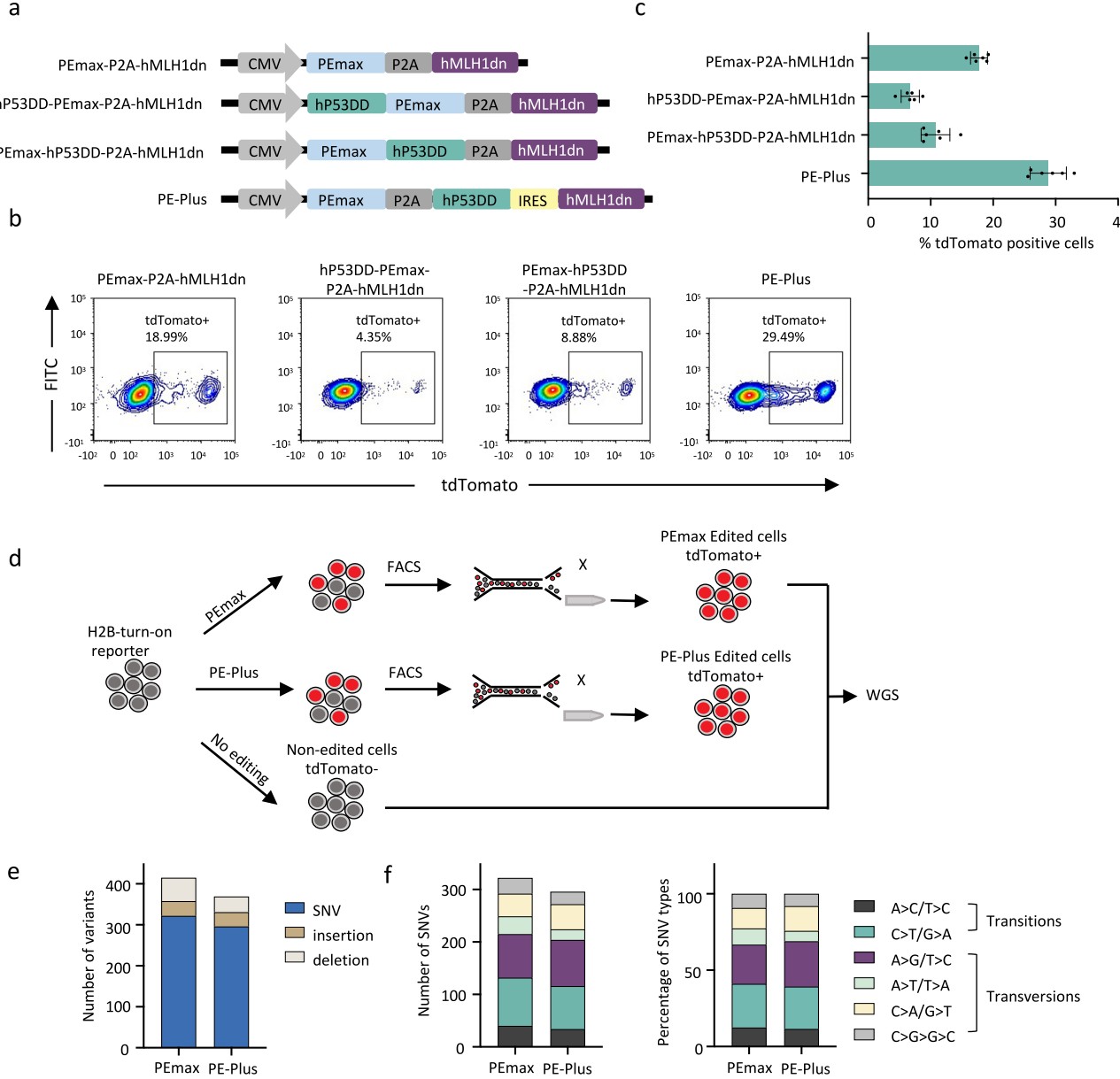

**Fig. 2 | Generation of all-in-one prime editor co-expressing PEmax, hMLH1dn, and hP53DD. a** Construction of all-in-one PEmax plasmid incorporating hMLH1dn and hP53DD simultaneously. The components are linked with PEmax via direct fusion or linkages, including P2A or IRES, as indicated. The PE-Plus plasmid consists of PEmax, hP53DD, and hMLH1dn linked with P2A and IRES in between. **b**, **c** Representative FACS plots (**b**) and bar graph (**c**) showing the proportions of tdTomato-positive cells at 48 h post-electroporation with the indicated prime editors together with the pegRNA in H2B-turn-on reporter cells. $n = 6$ independent electroporation reactions. Bars represent the mean ± S.D. **d** Schematic overview of experimental design to evaluate genome-wide off-target effects induced by PE-Plus and PEmax. The edited cells were isolated by FACS sorting of tdTomato-positive cells with frame restoration in the "H2B-turn-on reporter". SNVs and indels induced by these two prime editors were identified by comparing them to the unedited parental cells. **e** Number of SNVs, insertion, and deletions identified in the PEmax and PE-Plus edited cells. **f** Number of different types of SNVs (left panel) and their relative proportion (right panel) in the PEmax and PE-Plus edited cells. Source data are provided as a Source Data file for (**c**, **e**, and **f**).

To study whether inducible prime editing can also be temporally controlled during hPSC differentiation, we conducted neuroectoderm differentiation using the SOX2-H2B-turn-on reporter that carries the iPE-Plus platform. We generated neuron progenitor cells via dual SMAD inhibition[32] in 7 days, and maintained the NPCs for additional 7 days (Fig. 3f). Doxycycline was added to the cells during 7 days of Neuroectoderm induction or was added to NPCs from Day7 to Day 14 (Fig. 3f). Given that SOX2 is highly expressed in the neural progenitor cells (NPCs), the successful editing can be evaluated via tdTomato turn-on during the differentiation. Immunofluorescence staining demonstrated that a large population of cells at day 7 or day 14 of doxycycline treatment co-expressed PAX6 and tdTomato. Over 80% of tdTomato-

positive cells were detected upon doxycycline treatment at the two different stages (Fig. 3h, i) measured by FACS. This data showed the H2B editing could be robustly induced in the NPC differentiation process, as well as in hESC-derived NPCs. The editing efficiency in NPCs was similar to that in the undifferentiated hPSCs (Fig. 3d, e).

### Inducible disease-related mutations and evaluation

Next, we employed the iPE-Plus platform for the generation of disease-related mutations in hPSCs. In parallel, we created cells with inducible PE (iPE2) and inducible PEmax (iPEmax) (Supplementary Fig. 7). Two single clones were selected for each of the inducible cell lines and infected cells with epegRNA and sgRNA lentivirus designed to induce

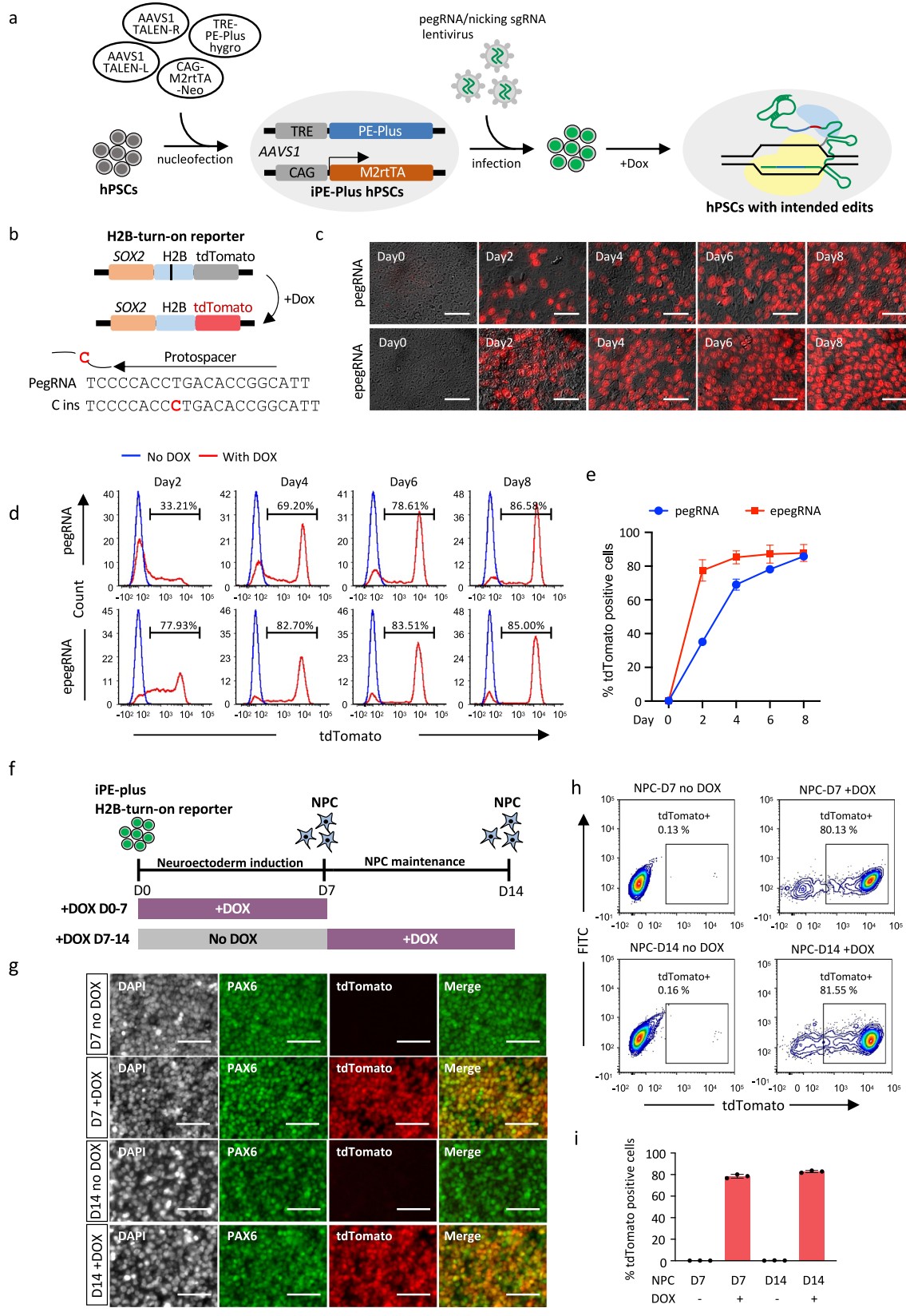

disease-related mutations: the *GBA* N370S mutation associated with Parkinson's disease (PD)[33] and *EGFR* L858R mutation linked to cancer[34]. To measure editing efficiency at various time points after doxycycline treatment, we utilized droplet digital PCR (ddPCR) to quantify mutation rates. In ddPCR, we designed a FAM-labeled probe to bind to the mutation sequence, while a HEX-labeled probe was designed to bind to

a non-targeted sequence within the same amplicon (Fig. 4a). The ratio of FAM positive to HEX positive events provides the mutation rate. In the absence of doxycycline, the mutation rates were undetectable, and all three inducible PE cell lines exhibited time-dependent induction of mutations as incubation time increased (Fig. 4b, c). Consistent with data generated by nucleofection, iPE-Plus demonstrated the highest editing

**Fig. 3 | Inducible prime editing in hPSCs with the iPE-Plus platform. a** Schematic workflow of iPE-Plus platform generation in hPSCs and induction of intended edits in the genome. **b** Schematic of doxycycline-inducible correction of a frameshift mutation in H2B with iPE-Plus platform in H2B-turn-on reporter. **c** Fluorescence images showing tdTomato activation using the iPE-Plus platform at indicated time points after doxycycline treatment. The iPE-Plus lines were transduced with either pegRNA (upper panel) or epegRNA (lower panel) lentivirus. Scale bar: 5 μm. Representative images from three independent experiments are shown. **d** Representative histograms at the indicated time points showing tdTomato-positive cell populations after doxycycline treatment in the presence of pegRNA (upper) or epegRNA (lower). Untreated (blue), doxycycline-treated (red). **e** Summary plot showing the average tdTomato percentage from three single-cell clones transduced with pegRNA and epegRNA lentivirus at indicated days of

doxycycline treatment. Data represent the mean ± S.D. from 3 independent experiments. **f** Schematic of inducible prime editing to correct frameshift mutations in H2B during neuroectoderm induction and maintenance. Doxycycline was added for 7 days at indicated stages. **g** Representative images from three independent experiments of immunofluorescence staining of PAX6 and co-expression with the tdTomato reporter gene in NPC cells after 7 days of doxycycline treatment during neuroectoderm induction or NPC maintenance. Scale bar: 5 μm. **h, i** Representative FACS plots (**h**) showing tdTomato-positive cells at day 7 or day 14 upon 7 days of doxycycline treatment. Corresponding cells without doxycycline treatment served as controls. The bar graph (**i**) depicts the mean percentage of edited cells ± S.D from 3 independent experiments. Source data are provided as a Source Data file for (**e** and **i**).

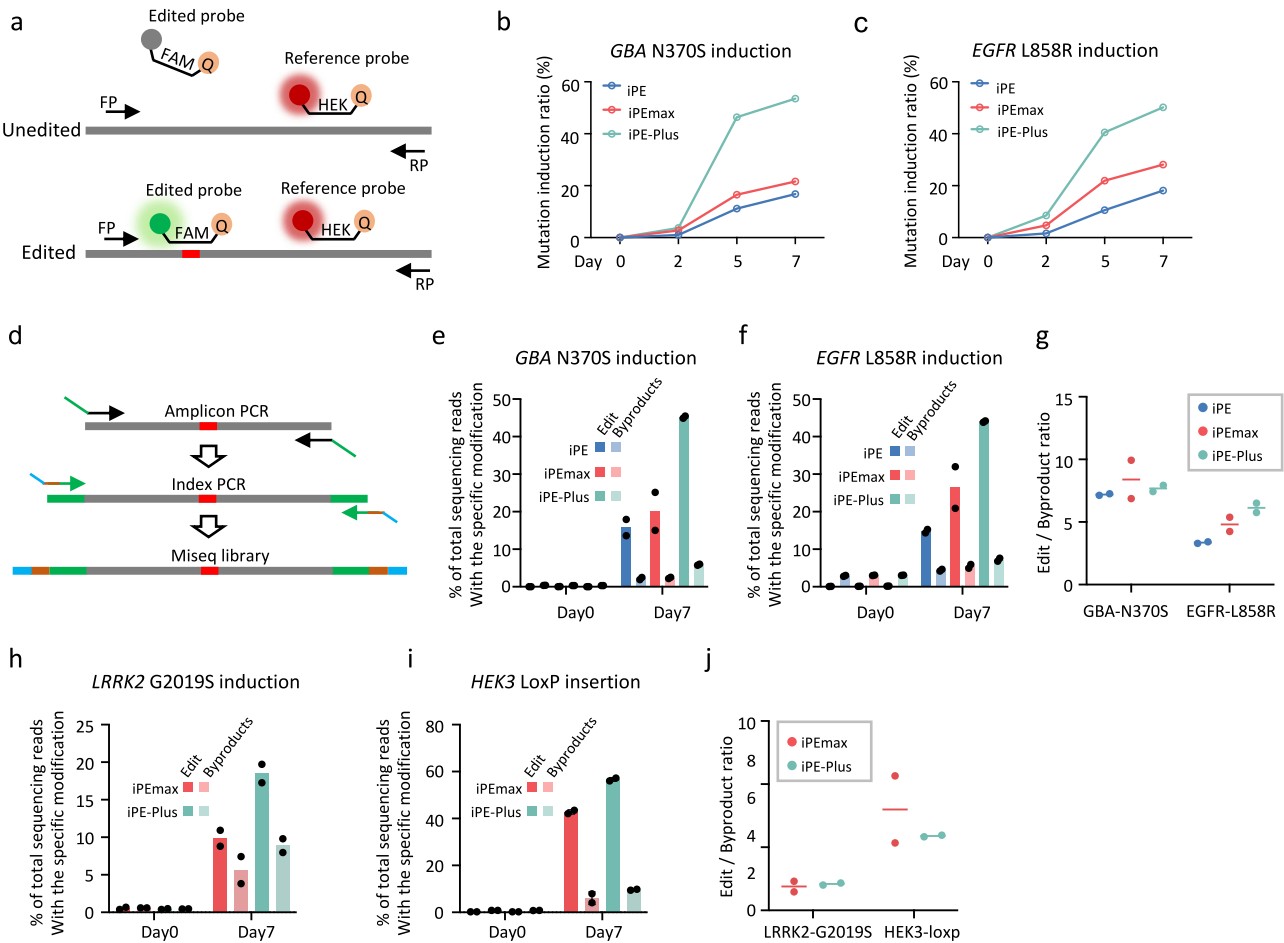

**Fig. 4 | Inducible installation of disease-related mutations in hPSCs. a** Schematic of quantifying mutation rates by ddPCR. **b, c** N370S mutation rate in *GBA* gene (**b**) and L858R mutation rate in *EGFR* gene (**c**) generated by iPE, iPEmax, or iPE-Plus platforms with different days of doxycycline induction, as determined by ddPCR. Data represents the mean from two single-cell clones for each type of inducible line. **d** Schematic of evaluating prime editing outcomes by Mi-seq. **e, f** Mi-seq analysis of intended editing and by-products of *GBA* N370S (**e**) and *EGFR* L858R (**f**) mutation induction using iPE, iPEmax, or iPE-Plus platforms before or after 7 days of doxycycline induction. Data represent the mean from two clones for each type of cell

line. **g** Evaluation of prime editing outcome purity by iPE, iPEmax, and iPE-Plus. Data are represented as the mean from two single-cell clones for each inducible line. **h, i** Miseq analysis of intended and unintended edits of the *LRRK2* G2019S mutation induction (**h**) and a "Loxp" insertion at the *HEK3* locus (**i**) using iPEmax or iPE-Plus. Bars represent the mean from two single-cell clones for each type of inducible line. **j** Prime editing purity calculated from h and i. Data are represented as the mean from two single-cell clones. Source data are provided as a Source Data file for (**b, c**, and **e–j**).

efficiency compared to iPE and iPEmax cells. After 7 days of doxycycline induction, iPEmax and iPE cells showed a *GBA* N370S mutation rate of 21.7% and 16.8%, respectively (Fig. 4b). and *EGFR* L858R mutation rates of 28.2% and 18.2%, respectively (Fig. 4c). In contrast, the desired mutation rates induced by iPE-Plus cells reached close to 50% for both *GBA* N370S and *EGFR* L858R mutations (Fig. 4b, c).

Editing frequencies from cells with 7 days of doxycycline induction were also measured by Miseq, and the editing efficiencies closely

matched the results from ddPCR (Fig. 4d–f). Meanwhile, the proportion of unwanted byproducts during prime editing was observed to slightly increase along with the intended edits. Nevertheless, the edit/byproduct ratio was similar between the three cell lines when inducing *GBA* N370S mutation (7.26 with iPE, 8.4 with iPEmax and 7.7 with iPE-Plus) and even higher for iPE-Plus when inducing the *EGFR* L858R mutation (3.4 with iPE, 4.8 with iPEmax and 6.1 with iPE-Plus) (Fig. 4g). The improvement of prime editing with iPE-Plus over PEmax was also

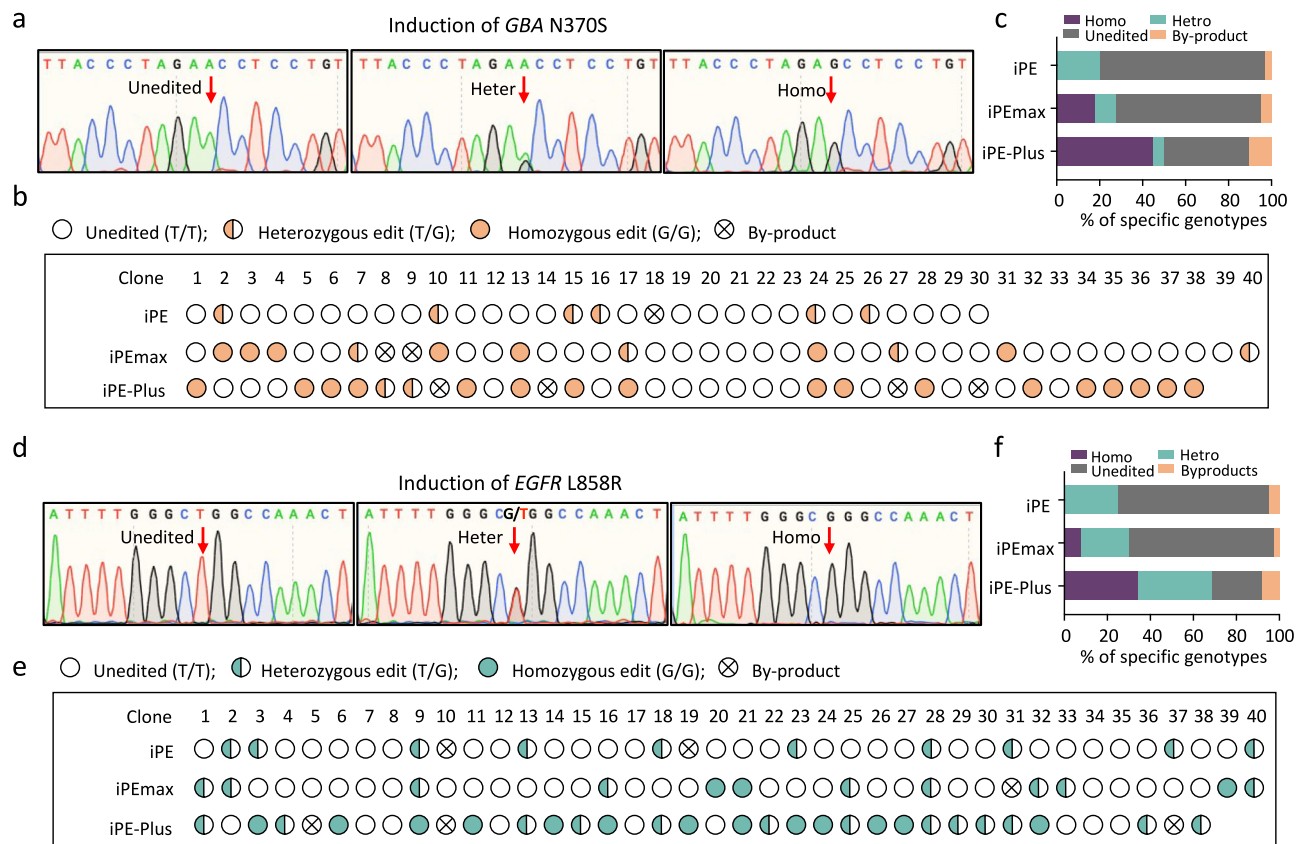

**Fig. 5 | Generation of single-cell clones carrying disease-related mutations using iPE, iPEmax and iPE-Plus platforms. a–c** Genotyping of single-cell clones from iPE, iPEmax, and iPE-Plus platforms with 7 days of induction of N370S mutation in *GBA*. Sequences of unedited, heterozygous, and homozygous mutated clones were determined by Sanger sequencing. The targeted locus is indicated by the red arrows (**a**). Genotyping for each clone is demonstrated with icons (**b**) and the proportions of different genotypes are summarized with a bar graph (**c**). **d–f** Genotyping of single-cell clones from the three platforms with 7 days of induction of *EGFR* L858R mutation via Sanger sequencing (**d**). Genotypes for single-cell clones were analyzed (**e**), and the percentages for each genotype were calculated (**f**) accordingly. Source data are provided as a Source Data file for (**c**, **f**).

observed when introducing another PD-related G2019S mutation at the *LRRK2* locus[35], increasing mutation rate from 9.9% to 18.5% (Fig. 4h), and a "Loxp" insertion at the *HEK3* locus, increasing from 42.8% to 56.7% (Fig. 4i), without sacrificing editing purity (Fig. 4j). We further applied iPE-Plus to induce *KRAS* mutations, the most common gene mutations related to cancer[36,37]. After 7 days of doxycycline induction, over 4% of G12C mutations and around 12% of G13A mutations were generated (Supplementary Fig. 8).

To further characterize induced prime editing in hPSCs, single-cell-derived clones were genotyped by Sanger sequencing from cells treated with doxycycline for 7 days. For *GBA* N370S mutation induction, 6 out of 30 clones from iPE cells were edited, all of which were heterozygous. iPEmax cells yielded 4 out of 40 clones with heterozygous edits and 7 out of 40 clones with homozygous edits. Among the 38 iPE-Plus clones, 2 were heterozygous, and 17 clones were homozygous (Fig. 5a–c). Similarly, for *EGFR* L858R mutation induction, 10 out of 40 cells had desired edits by iPE, all of which were heterozygous. iPEmax clones comprised 12 out of 40 edited clones, with 9 heterozygous and 3 homozygous. Among the 38 iPE-Plus clones, 13 were heterozygous, and 13 were homozygous (Fig. 5d–f). These data indicate that the proportion of edited clones increased from iPEmax cells compared with iPE cells, and further increased when using iPE-Plus cells, with a notable improvement in the efficiency of homozygous edits.

**Generation of multiplex disease-mutations in hPSCs**

Next, we investigated whether the iPE-Plus platform could be applied to achieve efficient one-step multiplexed prime editing. Our aim was to create a clinically relevant disease model in hPSCs by introducing two mutations at the *EGFR* locus: L858R and T790M mutations. The L858R mutation at the *EGFR* locus is one of the most common mutations in non-small-cell lung cancer (NSCLC) patients and can be targeted by EGFR tyrosine kinase inhibitors (EGFR-TKIs)[38]. However, the presence of the secondary T790M mutation leads to resistance to earlier generations of EGFR-TKIs, reducing median survival to less than 2 years after acquiring this mutation[39,40]. To induce the dual mutations simultaneously, we generated a lentivirus vector expressing dual pegRNAs and another vector expressing dual sgRNAs (Fig. 6a). Mutation rates for each *EGFR* mutation at different time points of doxycycline induction were measured by ddPCR and revealed a time-dependent accumulation of the two mutations (Fig. 6b). After 7 days of doxycycline induction, the mutation rates of L858R and T790M were approximately 20% and 25%, respectively (Fig. 6b, c). Different types of byproducts were observed to increase during the editing process, especially with the T790M mutation. This was also noted in the single-cell clone genotyping, where 6 out of 42 clones contained unwanted sequences, and 5 of them carried mutation near the targeting site for T790M induction (Fig. 6d). However, the higher byproduct rate was not attributed to multiplex PE editing, as the similar results were observed during single T790M mutation induction (Supplementary Fig. 9). Nevertheless, we successfully obtained single clones carrying different mutations with dox treatment, including monoallelic and biallelic single T790M mutations or single L858R mutations. Moreover, we obtained 5 clones with dual mutation, one with heterozygous L858R along with heterozygous T790M and four with heterozygous L858R along with homozygous T790M. These

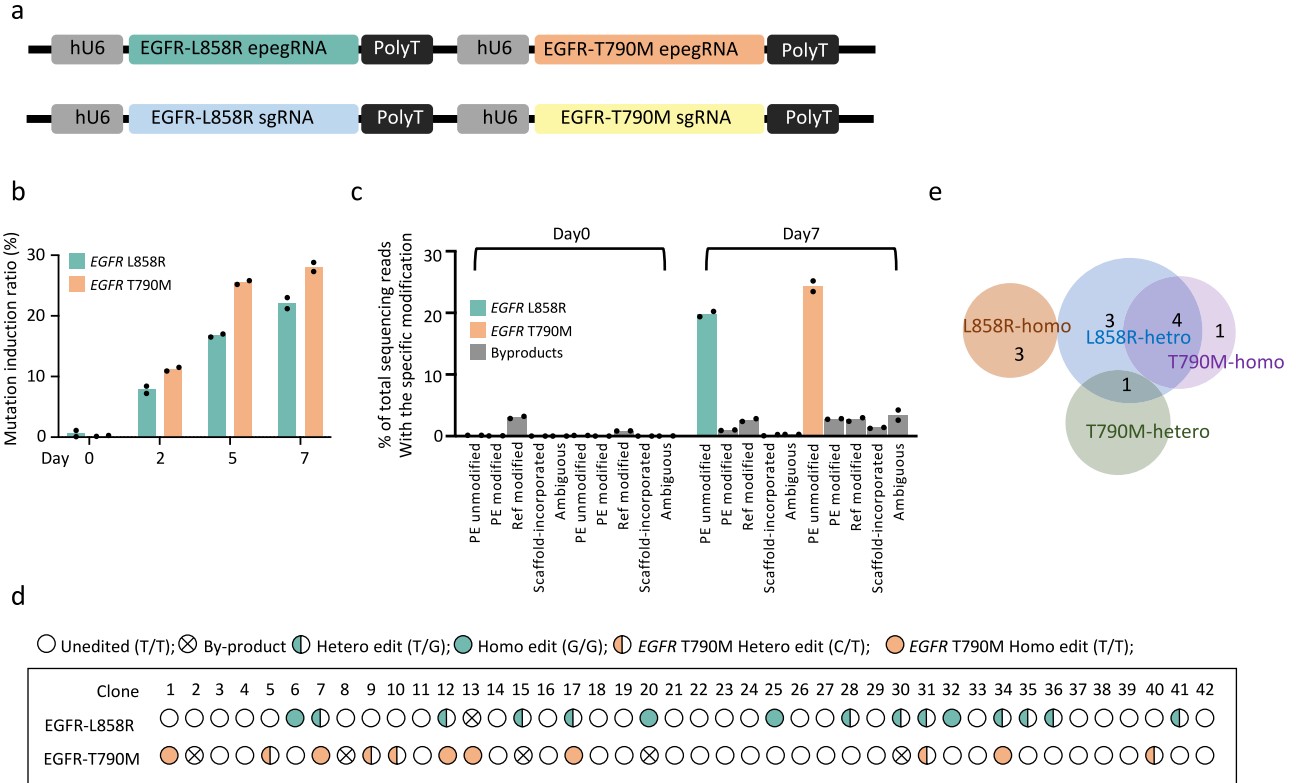

**Fig. 6 | One-step installation of multiplex mutations in hPSCs by the iPE-Plus platform. a** Construction of dual epegRNAs (upper) and dual nicking sgRNAs (lower) driven by tandem U6 promoters for the induction of L858R and T790M mutations. **b** The induction rate of two *EGFR* mutations by iPE-Plus platform at indicated time points, as determined by ddPCR. Data represents the mean from two iPE-Plus clones. **c** Miseq evaluation of the intended L858R and T790M mutations in single clones carrying multiple mutations offer a potential disease modeling platform for studying the mechanism of these mutations and evaluating new therapies.

*EGFR* as well as byproducts after 7 days of induction by the iPE-Plus platform. Bars represent mean from two iPE-Plus clones (**d**, **e**) Genotyping of single cell clone after 7-days of dual *EGFR* mutation induction using the iPE-Plus platform (**d**). Single clones with precise single mutations or double mutations were summarized with pie charts. Source data are provided as a Source Data file for (**b**, **c**).

## Discussion

The development of prime editing technology provides a powerful tool for generating hPSC-based disease models due to its precision and versatility[41,42]. However, achieving higher editing efficiencies remains a challenge. In this study, we conducted a comprehensive comparison of editing efficiency across a series of prime editors and developed a PE-Plus strategy by combining three key components- PEmax, MLH1dn and P53DD- that collectively contribute a very high editing efficiency. Previous studies have shown that inhibiting either the MMR or P53 pathway can improve PE efficiency[24,27]. In this study, we observe an additive effect when combining these two inhibitors with PE or PEmax (Fig. 1d, e). This additive effect aligns with the recently published paper by Park et al., where higher editing efficiency was achieved using the PE enzyme along with two pathway inhibitors[43]. Most importantly, we incorporated this PE-Plus system into the genome of hPSCs and demonstrated the applications of the iPE-Plus platform for inducible precise editing in hPSCs and their differentiated progeny. This system showcases high flexibility to install disease-relevant mutations in hPSCs and allows for the generation of monoallelic, biallelic, or even multiplex mutations via a one-step induction approach for stem cell research and disease modeling.

Our data showed that the effect of inhibiting the MMR pathway using MLH1dn was only observed in the smaller edits (Fig. 1d–f), but not in larger edits such as 10nt, 40nt deletion, and 34nt insertion tested in this study (Fig. 1k–n), indicating a size-dependent effect of

MMR inhibition in prime editing. This observation is consistent with previous studies from other groups showing the reduced impact of MMR on larger edits[24,44]. In contrast, PE-Plus remains effective for larger edits, and the improvement effects are mainly due to the P53DD component in the PE-Plus (Fig. 1k–n). Notably, the twin-PE mediated 40nt deletion was also improved by PE-Plus, indicating compatibility with the twin-PE system[28]. Combined with our previous data showing the improvement by MLH1dn and P53DD for point mutations and 1nt insertion editing, our study showed the PE-Plus system is effective for both smaller and larger edits in hPSCs.

To perform prime editing in hPSCs, multiple plasmids, including prime editor, pegRNA, and nicking sgRNA are transiently transfected into the cells in a simultaneous manner. However, the quality of plasmid DNA, the number of plasmids, the size of the plasmids, cell density, and transfection methods can all impact transfection efficiency and lead to cellular toxicity[45,46]. This, in turn, results in low efficiency and reduced reproducibility of the prime editing. Therefore, we developed a universal platform for prime editing in hPSCs called "iPE-Plus". With this platform, prime editing is induced by doxycycline, eliminating the side effects caused by variations in transfection efficiency, reducing cellular toxicity associated with the transfection procedure, and ensuring the reproducibility of prime editing. It is worth noting that in the absence of doxycycline, the prime editor expression was extremely low (Supplementary Fig. 5c), and the editing events were barely detected (Fig. 4e, f, h, i). This indicates minimal "leakiness" in this platform, making it a reliable isogenic control for the study of intended mutations.

The iPE-Plus platform enables temporal control of prime editing, allowing editing events to accumulate with the prolonged duration of doxycycline treatment. This is evident in the kinetics of genome

editing observed in fluorescence reporter lines (Fig. 3b–e and Supplementary Fig. 6), as well as in the generation of disease-related mutation generation (Fig. 4b, c). This feature enhances the efficiency and feasibility of creating hPSC-based disease models with specific mutations through prime editing. Since the prime editing efficiency can vary depending on the case, the duration of doxycycline treatment can be adjusted to achieve an optimized editing ratio. Another benefit of temporally inducing precise mutations is the ability to introduce desired mutations at specific developmental stages or only within certain cell types. In this study, we demonstrated that the iPE-Plus system is capable of introducing desired edits in different cell types, including in undifferentiated hPSCs, in hPSCs undergoing neuroectoderm induction, and in PAX6-positive NPCs. Since some gene mutations lead to pleiotropic effects during hPSC differentiation, the iPE-Plus platform provides more accurate models for elucidating and dissecting the pathological role of disease-related mutations. In addition to iPE-Plus, we have also developed iPE and iPEmax platforms for side-by-side comparisons of the editing efficiency. While these two systems achieved a lower editing efficiency compared to iPE-Plus, researchers can choose to use them for their specific purpose, especially if concerns arise regarding the prolonged expression of the two pathway inhibitors MLH1dn and P53DD.

Based on the genotyping results from hPSC single-cell clones, we observed that in addition to the overall improvement in editing efficiency achieved with iPE-Plus, there was also an increase in clones displaying homozygous editing. With iPE-Plus, both monoallelic and biallelic models can be obtained in single-step induction (Fig. 5). The cell model carrying homozygous mutations demonstrates stable phenotypes, which is essential for studying disease-related mutations without producing trait segregation. Furthermore, due to this capability, iPE-Plus has the potential to generate knockout cell models for studying gene function by precisely introducing premature stop codons into targeted genes (Supplementary Fig. 6). The traditional method for generating gene knockout models in hPSC involves the use of wild-type Cas9 to induce DSBs, which can lead to cellular toxicities due to DNA damage, compromised genome integrity, and unpredictable editing sequences[47–50]. In contrast, the iPE-Plus platform offers a DSB-independent genome editing approach, introducing premature stop codons at desired sites, and achieving precise gene knockouts.

Finally, we demonstrated the ability of the iPE-Plus platform to induce multiple mutations. While some genetic diseases are caused by single mutations, most diseases represent complex disorders associated with risk derived from complex genetic mutations. For instance, cancer often exhibits a multitude of mutations, and the increasing complexity of these mutations makes cancer therapy challenging[51,52]. There is an urgent need for robust genetic models to study both single and multiple mutations contributing to tumor progression and treatment resistance. The iPE-Plus provides a universal platform for investigating multiple mutations within the same genetic background. In our study, we successfully generated cell models carrying both *EGFR* L858R and T790M mutations (Fig. 6), which represent a molecular feature of drug resistance to early generations of EGFR-TKI treatment. With this platform, single-cell clones carrying either a single mutation or dual mutations can be easily obtained in a single step without the need for additional rounds of genome editing. The platform provides an approach to generate robust models in hPSCs, which can be propagated and differentiated into specific cell types for studying the complex mutation interactions in various diseases.

## Methods
### Culture conditions for hPSCs
All the hPSCs are cultured on Matrigel (Fisher Scientific 08-774-552) in Stemflex Medium (Thermo Fisher A3349401) and fed daily. Cells were passaged with a ratio of 1:4 when they reached 70%–80% confluency by incubation cells with 0.5 mM EDTA (Fisher Scientific MT-46034CI) at room temperature for 5 min.

### Construction of plasmids
To generate a plasmid expressing human P53DD under the EF1α promoter, a DNA gblock containing the C-terminal region of human P53, which includes the NLS, TET, and CTD domains, was cloned into the pEF-GFP plasmid (Addgene #11154) by replacing the GFP. Two truncated P53DD, one containing containing only the TET and CDT domains and another with only the TET domain were amplified from the full-size hP53DD and cloned into pEF-GFP using the same strategy.

To generate all-in-one prime editors with hP53DD directly fused with PEmax, the hP53DD fragment was cloned into PEmax-P2A-hMLH1dn (Addgene #174828) at the C-terminus (hP53DD-PEmax-P2A-hMLH1dn) or N-terminus of PEmax (PEmax-hP53DD-P2A-hMLH1dn) under the constitutive expression from a CMV promoter using Hifi DNA assembly (NEB). To generate the other two all-in-one prime editors with linkages between the three components, the BSD fragment in pCMV-PEmax-P2A-BSD (Addgene # 174821) was first replaced with the hP53DD fragment to generate pCMV-PEmax-hP53DD. The IRES -hMLH1dn fragments or P2A-hMLH1dn were then cloned into the 3' end of hP53DD to generate pCMV-P2A-hP53DD-IRES2-hMLH1dn (PE-Plus) and pCMV-P2A-hP53DD-P2A-hMLH1dn.

The donor plasmids for the inducible expression of PE2, PEmax or PE-Plus were constructed by replacing Cas9 in the Hygro-Cas9 donor plasmid (Addgene #86883) with the respective PE2, PEmax or PE-Plus fragments, which were amplified from pCMV-PE2 (Addgene #132775), pCMV-PEmax-P2A-BSD (Addgene # 174821) or the cloned PE-Plus.

For the construction of empty pegRNA and epegRNA lentivirus backbones, the lenti sgRNA(MS2)_puro plasmid (Addgene #73797) was linearized with BamHI and NdeI. The fragment containing the RFP dropout cassette and the fragment containing the RFP cassette along with the mpknot motif were separately amplified from pU6-pegRNA-GG-aceptor (Addgene #132777) and pU6-tmpknot-GG-acceptor (Addgene #174039). These fragments were then assembled with the linearized lentivirus vector to generate the lenti-pegRNA-GG-acceptor and lenti-tmpknot-GG-aceptor backbone plasmids under the hU6 promoter. Two additional BsmbI sites were introduced into these backbone plasmids during PCR amplification for cloning of the designed pegRNA sequence.

The cloning of plasmids expressing pegRNAs or nicking sgRNAs was performed as described in a published protocol[41]. Briefly, the pegRNA backbone plasmid was linearized with enzyme digestion: pU6-pegRNA-GG-acceptor plasmid, pU6-tevopreq1-GG-acceptor (Addgene #174038) or pU6-tmpknot-GG-acceptor was digested with BsaI-HFv2 (NEB), while lenti-pegRNA-GG-acceptor or lenti-tmpknot-GG-acceptor was digested with BsmBI-v2 (NEB). The pegRNA or epegRNA plasmid was cloned using Golden-Gate assembly, assembling linearized backbone, pegRNA spacer sequence, pegRNA scaffold, and pegRNA extension sequence. Nicking sgRNA for PE3 editing was cloned into the LsgRNA backbone (Addgene #47108) through the BbsI site or into lenti-sgRNA blast (Addgene #104993) through the BsmbI site. A list of pegRNAs and nicking sgRNAs used in this work is provided in Supplementary Table 1. To generate a single epegRNA plasmid or nicking sgRNA with tandem hU6-driven epegRNAs or nicking sgRNAs for the simultaneous induction of two *EGFR* mutations, the fragment containing hU6 promoter, epegRNA or sgRNA for T790M mutation, and the PolyT sequence was cloned into the 3' end of PolyT sequence of the lenti-epegRNA or lenti-sgRNA for *EGFR* L858R using Hifi DNA assembly.

### hPSC lines
H1 hESCs were purchased from WiCell Institute. The H1-SOX2-tdTomato reporter line was generated by knocking in the P2A-H2B-tdTomato cassette before the stop codon at the *SOX2* locus through CRISPR-mediated HDR in H1 cells. The H2B-tdTomato turn-on reporter

line was generated by knocking in a mutated P2A-H2B-tdTomato cassette before the stop codon at the *SOX2* locus through CRISPR-mediated HDR in H1 cells. The mutated P2A-H2B-tdTomato cassette carried a "C" deletion in the H2B sequence. Both reporter lines harbored heterozygous insertions of the cassette. The iPE2, iPEmax and iPE-Plus lines with inducible expression of the three prime editors were generated by introducing the donor plasmids containing the prime editor under the TRE-tight promoter, the Neo-M2rtTA donor (Addgene #60843) and a pair of TALENs (Addgene # 59025, # 59026) through co-electroporation into hPSCs, targeting the first intron of the *PPP1R12C* gene[53]. Single cells were isolated in 96-well plates and subjected to double selection with G418 and hygromycin. Positive clones were subsequently confirmed through PCR and Sanger sequencing (Supplementary Table 4). For inducing prime editing experiments in all the inducible PE lines, doxycycline at a working concentration of 2 μg/mL (Sigma-Aldrich) was applied.

### Electroporation of plasmids in hPSCs

hPSCs were dissociated using Accutase (Innovative Cell Tech. AT104) at 37 °C for 10 min, and 250,000 single cells were used for a small reaction of electroporation (Lonza V4XP-3032) following the manufacturer's instructions. The reactions were performed using the "CB-150" program on the Lonza 4D-Nucleofector X Unit. The cells from one reaction were subsequently seeded into a single well of a Matrigel-coated 24-well plate.

### Whole genome sequencing and data analysis

Genomic DNA from edited and wild-type hPSCs was isolated using DNeasy Blood & Tissue Kit (QIAGEN) according to the manufacturer's instructions. After PicoGreen quantification and quality control by Agilent TapeStation, 257–287 ng of genomic DNA were sheared using a LE220-plus Focused-ultrasonicator (Covaris catalog # 500569), and sequencing libraries were prepared using the KAPA Hyper Prep Kit (Kapa Biosystems KK8504) with modifications. Briefly, libraries were subjected to a 0.5X size selection using aMPure XP beads (Beckman Coulter catalog # A63882) after post-ligation cleanup. Libraries were amplified with 5 cycles of PCR and pooled equimolar.

Samples were run on a NovaSeq X in a PE150 run, using the NovaSeq X 25B Reagent Kit (Illumina). The average number of read pairs per sample was 506 million, corresponding to 39X mean coverage. The whole-genome sequencing data underwent processing using the Illumina BaseSpace DRAGEN Somatic application v4.2.4 with default configurations. Initially, the sequencing reads were aligned to the human GRCh38 reference genome. Subsequently, duplicate aligned reads were identified and removed from downstream analysis. The edited samples were then compared against the wild-type sample to detect single nucleotide variants (SNVs) and small insertions/deletions (indels).

### Lentivirus production and transduction

HEK293T cells were cultured with high-glucose DMEM supplemented with 10% fetal bovine serum in 10 cm Petri dishes. When HEK293T cells each had 95% confluence, cells were split into 10 cm dishes at a 1:5 ratio to achieve 70% ~ 80% confluency at the time of transfection. To package lentivirus, a mixture containing 10 μg psPAX6, 6 μg PMD2.G, and 10 μg of the lentiviral plasmid of interest was co-transfected into HEK293T cells in the presence of 60 μL 1 mg/mL PEI. After transfection for 12 h ~ 16 h, the culture medium was replaced with a fresh HEK293T culture medium. Viral supernatants were collected 48 h after transfection, followed by centrifugation at 500 × *g* for 5 min and filtration through a 0.45 μm PVDF filter (Corning) to remove cellular debris.

To transduce cells with pegRNA, epegRNA, or nicking sgRNA lentivirus, iPE, iPEmax, or iPE-Plus cells were passaged to reach around 30% ~ 40% confluence on the next day in a 6-well plate. For the H2B-turn-on-reporter or H1-SOX2-tdTomato reporter expressing iPE-Plus,

500 μL of pegRNA or epegRNA was added to the cells along with 10 μg/ml polybrene. Fresh Stemflex medium was changed on the next day, and Stemflex containing 1 μL/mL puromycin was applied to the cells for selection 48 h after transduction for 3 days. For all other endogenous gene editing, including induction disease-related mutations, insertions, and deletions, 500 μL epegRNA, as well as 500 μL nicking sgRNA lenvirus were added to the cells simultaneously. Dual drug selection was conducted 48 h after transduction using 1 μL/mL puromycin plus 10 μg/ml blasticidin for 3 days.

### Evaluation of prime editing efficiency by flow cytometry

Reporter lines were dissociated into single cells by incubating cells in Accutase (Innovative Cell Tech. AT104) at 37 °C for 10 min. Accutase was removed by centrifugation, and the cell pellets were resuspended in cold PBS containing 0.5% BSA and filtered through a cell strainer with 35 μm sized mesh (Fisher Scientific 352235) to remove clumps. Cells expressing tdTomato were analyzed using a BD FACSAria III (BD Bioscience). Data analysis was conducted with FCS Express software (version 7.18.0025, DeNovo Software). The gating strategy is shown in Supplementary Fig. 10. The GFP channel was used as a no-fluorescence control only for gating of tdTomato-positive or tdTomato-negative population. PE efficiency was measured in the H2B-turn-on reporter line, where the percentage of tdTomato-positive cells reflected cells with editing. For the endogenous TGA insertion in the H1-SOX2-tdTomato line, the editing efficiency was measured with tdTomato-negative cells.

### Evaluation of prime editing efficiency by droplet digital PCR

Disease-related mutation rates induced by prime editing were quantified using droplet digital PCR (ddPCR). Genomic DNA was extracted using QuikExtract DNA Extraction Solution (Epicenter). A pair of primers was designed to amplify the region spanning the targeted site. A FAM-labeled probe was designed to bind the intended mutation, and a HEX-labeled probe was designed to bind a non-targeted region but within the same amplicon (Fig. 4a). Droplets were generated using a QX200 Manual Droplet Generator (Bio-Rad) according to the manufacturer's instructions. The primer and probe sequence for each mutation were listed in the Supplementary Table 2. Droplets were read on a QX200 Droplet Reader (Bio-Rad) and analyzed using QuantaSoft (version 1.4, Bio-Rad). The mutation rate was calculated as the ratio of FAM-positive to HEX-positive droplets. Representative 2D plots are shown in Supplementary Fig. 11.

### Evaluation of prime editing outcomes by Miseq

Cells with or without mutation induction were lysed using Solution for Blood (Millipore Sigma L3289) and then neutralized with Neutralization Solution for Blood (Millipore Sigma SRE0087). The editing region was amplified with PCR using Q5 High-Fidelity 2X Master Mix (NEB) followed by purification the PCR product using the QIAquick PCR Purification Kit (QIAGEN). Primers for PCR were listed in Supplementary Table 3. The purified amplicons were submitted for amplicon sequencing (Illumina MiSeq system, Amplicon EZ service from Genewiz). The fastq files were analyzed using CRISPResso2 with the following parameters: a minimum homology for the alignment to an amilicon > 60%; a minimum average read quality (phred33 scale) > 30; and exclusion of 15 bp from the left side and 15 bp from the right side of the amplicon sequence for the quantification of the mutations. HDR mode was used for quantification of the desired 40nt deletion at the *SOX2* locus by TWIN-PE. The two pegRNA targets were provided, and the quantification window was set to -3. The editing efficiency was calculated as a percentage of perfect HDR-aligned reads/total aligned reads. For the rest of the prime editing cases using single pegRNA, the data were analyzed using the prime editing model. The frequencies of intended edits and different byproducts including indels and scaffold incorporation, were determined.

## Genotyping of single-cell clones

Intended mutations were induced in iPE-Plus cells with doxycycline treatment for 7 days, after which single cells were split into 96-well plates. Individual single-cell clones were picked for 2 weeks and lysed for PCR using the same primers as for the Miseq amplicons. The PCR products were purified with a QIAquick PCR Purification Kit (QIAGEN) and then submitted for Sanger sequencing. Primers for PCR amplification and sequencing are listed in Supplementary Table 4.

## Neuroectoderm induction and NPC maintenance

Neuroectoderm induction was performed using the dual SMAD inhibition method with SB431542 (Cayman Chemicals) and LDN193189 (Reprocell) for treatment. Briefly, 400,000 undifferentiated hPSCs were resuspended in Essential 6 medium (Gibco) supplemented with 500 nM LDN193189 and 10 μM SB431542. The cells were then seeded onto a Matrigel-coated 24-well plate and cultured with daily feeding for 7 days. The NPC cells at day 7 were maintained in the same well for an additional 7 days. Doxycycline (2 μg/mL, Sigma-Aldrich) was added to the cells from day 0 to day 7 or from day 7 to day 14, followed by immunostaining and FACS analysis.

## Immunofluorescence staining

The NPC cells at day 7 or day 14 with doxycycline treatment were fixed with 4% paraformaldehyde for 15 min at room temperature, followed by three washes with PBS. Subsequently, the fixed cells were incubated in a blocking solution containing 5% goat serum (Gibco) and 0.3% Triton™ X-100 (Sigma-Aldrich) in PBS for 1 h at room temperature. The cells were then incubated with a PAX6 antibody (1:200, 561462, BD) diluted with PBS plus 1% Goat Serum and 0.15 % Triton™ X-100 at 4 °C overnight. After three washes with PBS, the cells were incubated with a goat anti-mouse secondary antibody conjugated with Alexa Fluor 488 (1:500, Thermo Fisher Scientific) for 1 h at room temperature. Following three PBS washes, the cells were incubated in DAPI for nuclei staining (1 μg/mL, Thermo Scientific) for 10 min. After three more PBS washes, Images were captured by a fluorescence microscope (Olympus IX81).

## Statistics & reproducibility

The statistical analyses were performed using GraphPad Prism 9. Statistical significance was calculated by ordinary one-way ANOVA or Student's two-tailed $t$ test as described in each figure legend. The sample size for each experiment and $p$-value were provided.

## Reporting summary

Further information on research design is available in the Nature Portfolio Reporting Summary linked to this article.

## Data availability

Miseq, sanger sequencing, and whole genome sequencing data are deposited in the NCBI BioProject accession code PRJNA1047080. The plasmids generated in this study have been deposited in Addgene [https://www.addgene.org/browse/article/28243876/]. All the data have been made available by deposition or within the source data. Source data are provided in this paper.

## Code availability

The Benchling CRISPR Design website was used for design sgRNA or pegRNA spacer and is available at [https://www.benchling.com/#]. CRISPResso2 website was used for analysis of Amplicon sequencing (Miseq) to determine the prime editing efficiency and is available at [https://crispresso.pinellolab.partners.org/submission].

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

## Acknowledgements

This work was supported by NIH/NCI Cancer Center Support Grant P30 CA008748 from Memorial Sloan Kettering to Stem Cell Research Facility. The work was also supported by a core facility grant of the Starr Foundation through the Tri-Institutional Stem Cell Initiative and by Contract C029153 from the New York State's stem cell funding agency (NYSTEM) to L.S. We acknowledge the use of the Integrated Genomics Operation Core, funded by the NCI Cancer Center Support Grant (CCSG, P30 CA08748), Cycle for Survival, and the Marie-Josée and Henry R. Kravis Center for Molecular Oncology. We thank Dr. Tuo Zhang at the Weill Cornell Genomics Resources Core Facility for processing the WGS data.

## Author contributions

Y.W., T.Z., and L.S. conceived the project and designed the experiments. Y.W., A.Z., M.S., T.W.K., B.R., B.P., and T.Z. performed experiments and analyzed data. Y.W., T.Z., and L.S. wrote the manuscript.

## Competing interests

L.S. is a scientific co-founder and consultant of BlueRock Therapeutics. The other authors have no competing interests.
