## [Transparent Peer Review file · Nature Communications]

Robust and inducible genome editing via an all-in-one prime editor in human pluripotent stem cells

Corresponding Author: Dr Ting Zhou

Version 0:

Reviewer comments:

Reviewer #1

(Remarks to the Author)

The authors responded to most of the issues I raised in the last review, and I felt that the revised edition was greatly improved. Recently, however, a paper demonstrating similar strategy in iPSC has been published in Nature Communications [Park et al., Nature Communications 15: 4002 (2024)]. It is necessary to clarify differences or superiority against the study.

Reviewer #2

(Remarks to the Author)

The authors have addressed my previous concerns; no additional experiments are needed.

Reviewer #3

(Remarks to the Author)

The authors have addressed my concerns about the clarity of the manuscript. The issue about tdTomato activation being by random deletion has also been covered by the additional sequencing data.

We greatly appreciate all the reviewers' carefully review and positive feedback.

REVIEWERS' COMMENTS

Reviewer #1 (Remarks to the Author):

The authors responded to most of the issues I raised in the last review, and I felt that the revised edition was greatly improved. Recently, however, a paper demonstrating similar strategy in iPSC has been published in Nature Communications [Park et al., Nature Communications 15: 4002 (2024)]. It is necessary to clarify differences or superiority against the study.

We appreciate the reviewer's comments. We noticed the paper by Park et al. during our paper revision. Their study demonstrated an improvement in the editing efficiency of base editing and prime editing by simultaneously inhibiting p53 and MMR pathways in hPSCs. We agree with the reviewer it is important to clarify the differences and the superiority of our study.

We have now referenced this paper, and included a discussion in the revised manuscript, as below:

"This additive effect aligns with the recently published paper by Park et al., where higher editing efficiency was achieved using the PE enzyme along with two pathway inhibitors⁴³. Most importantly, we incorporated this PE-Plus system into the genome of hPSCs and demonstrated the applications of the iPE-Plus platform for inducible precise editing in hPSCs and their differentiated progeny. This system showcases high flexibility to install disease-relevant mutations in hPSCs and allows for the generation of monoallelic, biallelic, or even multiplex mutations via a one-step induction approach for stem cell research and disease modeling."

Reviewer #2 (Remarks to the Author):

The authors have addressed my previous concerns; no additional experiments are needed.

Reviewer #3 (Remarks to the Author):

The authors have addressed my concerns about the clarity of the manuscript. The issue about tdTomato activation being by random deletion has also been covered by the additional sequencing data.